# Maternal Obesity Programs the Premature Aging of Rat Offspring Liver Mitochondrial Electron Transport Chain Genes in a Sex-Dependent Manner

**DOI:** 10.3390/biology12091166

**Published:** 2023-08-24

**Authors:** Consuelo Lomas-Soria, Guadalupe L. Rodríguez-González, Carlos A. Ibáñez, Luis A. Reyes-Castro, Peter W. Nathanielsz, Elena Zambrano

**Affiliations:** 1Departamento de Biología de la Reproducción, Instituto Nacional de Ciencias Médicas y Nutrición Salvador Zubirán, Mexico City 14080, Mexico; mconsuelo@conahcyt.mx (C.L.-S.); guadalupe.rodriguezg@incmnsz.mx (G.L.R.-G.); carlos_albertoibc@hotmail.com (C.A.I.); luis.reyesc@incmnsz.mx (L.A.R.-C.); 2CONAHCyT-Cátedras, Investigador por México, Departamento de Biología de la Reproducción, Instituto Nacional de Ciencias Médicas y Nutrición Salvador Zubirán, Mexico City 14080, Mexico; 3Wyoming Center for Pregnancy and Life Course Health Research, Department of Animal Science, University of Wyoming, Laramie, WY 82071, USA; peter.nathanielsz@uwyo.edu; 4Department of Genetics, Texas Biomedical Research Institute, San Antonio, TX 78227, USA; 5Facultad de Química, Universidad Nacional Autónoma de México, Mexico City 04510, Mexico

**Keywords:** maternal obesity, aging, RNA-seq, liver, mitochondria, oxidative phosphorylation

## Abstract

**Simple Summary:**

Developmental programming is now an area of considerable interest throughout the biomedical research community as it is now well accepted that challenges during fetal and early neonatal life program the trajectory of the development and function of multiple systems across the life span. There is now also compelling evidence that developmental programming alters the trajectory of aging, beginning early in life. The present study links mitochondrial function to molecular signaling pathways that regulate life span and to the aging process; it demonstrates the role and importance of mitochondria in the predisposition to developing a fatty liver. The overall message we wish to emphasize is that hepatic aging in offspring caused by maternal obesity in rats involves changes in the mitochondrial function pathways that result in fatty livers. These processes show sexual dimorphism as they occur in males and females at different ages. These findings throw new light on the mechanisms that underlie the well-established sexual dimorphism in aging. We hope this paper will be a stimulus to similar studies on other tissues.

**Abstract:**

We investigated whether maternal obesity affects the hepatic mitochondrial electron transport chain (ETC), sirtuins, and antioxidant enzymes in young (110 postnatal days (PND)) and old (650PND) male and female offspring in a sex- and age-related manner. Female Wistar rats ate a control (C) or high-fat (MO) diet from weaning, through pregnancy and lactation. After weaning, the offspring ate the C diet and were euthanized at 110 and 650PND. The livers were collected for RNA-seq and immunohistochemistry. Male offspring livers had more differentially expressed genes (DEGs) down-regulated by both MO and natural aging than females. C-650PND vs. C-110PND and MO-110PND vs. C-110PND comparisons revealed 1477 DEGs in common for males (premature aging by MO) and 35 DEGs for females. Analysis to identify KEGG pathways enriched from genes in common showed changes in 511 and 3 KEGG pathways in the male and female livers, respectively. Mitochondrial function pathways showed ETC-related gene down-regulation. All ETC complexes, *sirtuin2*, *sirtuin3*, *sod-1*, and *catalase*, exhibited gene down-regulation and decreased protein expression at young and old ages in MO males vs. C males; meanwhile, MO females down-regulated only at 650PND. Conclusions: MO accelerates the age-associated down-regulation of ETC pathway gene expression in male offspring livers, thereby causing sex-dependent oxidative stress, premature aging, and metabolic dysfunction.

## 1. Introduction

Obesity is considered a global pandemic and a worldwide public health issue [1]. In recent decades, the proportion of women of reproductive age who are overweight and obese has increased significantly, as has the incidence of obesity during pregnancy [2,3]. Human and controlled animal studies have shown that maternal obesity has detrimental lifelong consequences on offspring by programming their cells, tissues, and organs, as well as their structures and functions. Maternal obesity programs offspring metabolic disorders through several mechanisms, including metabolic, hormonal [4,5,6,7], and epigenetic changes [8,9], as well as through oxidative stress [10,11], a commonly proposed mechanism for liver injury and the progression of age-related diseases [10]. Non-alcoholic fatty liver disease (NAFLD) is the most prevalent form of liver disorder and is considered a global epidemic. Obese young children and adults are more likely to develop early liver diseases [12,13].

There is evidence that the onset of a fatty liver in offspring may occur early in development [14,15]. Moreover, maternal obesity accelerates the onset of offspring metabolic and liver dysfunction and shortens life span [10,16,17].

We have reported that maternal obesity programs premature metabolic aging in offspring in a sex-dependent manner, possibly due to increased oxidative stress, changes in steroid hormones, cardiovascular changes, and other functional alterations [10]. In most cases of programming by maternal obesity, such as NAFLD, we, and others, have observed a sexual dimorphism of outcomes as the male offspring of obese mothers exhibit more pronounced NAFLD characteristics (physiological, biochemical, histological, and gene changes) than the females [4,10].

Aging is a complex and dynamic biological process that, over time, causes a variety of structural, functional, molecular, and cellular damage, thereby increasing the risk of multiple diseases [18]. Cellular senescence, including telomere shortening and genomic and mitochondrial DNA damage, is a major cause of aging, and plays an important role in the progression of NAFLD and other liver diseases [19,20,21]. Mitochondrial dysfunction, such as a decreased oxidative capacity and increased reactive oxygen species (ROS) production, has been proposed as a cellular and molecular hallmark of aging [22,23].

Mitochondria are highly sensitive to their environmental conditions and undergo adaptations during the development of NAFLD diseases. Thus, mitochondrial alterations are implicated in liver aging and fatty liver diseases [24,25,26,27,28,29,30]. Mitochondrial dysfunction has been proposed as a central process in the development of liver disease programmed by maternal obesity [31]. Offspring of obese rodent mothers exhibit elevated oxidative stress and mitochondrial dysfunction in both fetal and young adult livers [32,33,34,35]. Some studies have evaluated the effects of maternal obesity programming on mitochondrial gene expression in offspring livers [15,36,37,38]. However, few studies have examined the extent to which these changes persist until old age. There is a need to determine whether adverse outcomes can be caused by aging and/or by maternal diet.

In the current study, RNA-seq was used to determine the difference in transcriptome changes between the male and female offspring of obese mothers at the young age of 110 and the old age of 650PND. We also measured protein products for key genes with altered expressions. We focused on the liver mitochondrial oxidative phosphorylation pathway in the young and old male and female offspring of control and obese mothers. Importantly, in our rat colony, the offspring of the control mothers (normally fed and normal weight) lived for ~850 days; whereas, the offspring of the obese mothers did not live for much longer than 650PND. 

We hypothesized that maternal obesity: (1) produces sex-specific age-related liver transcriptome changes in offspring; (2) leads to offspring liver dysfunction by impairing antioxidant defenses and mitochondrial function; and (3) causes offspring hepatic mitochondrial electron transport chain (ETC) gene down-regulation, leading to an increase in reactive oxygen species (ROS) concentrations and liver damage.

## 2. Materials and Methods

### 2.1. Animals

The Animal Experimentation Ethics Committee of the Instituto Nacional de Ciencias Médicas y Nutrición Salvador Zubirán (INCMNSZ), Mexico City, Mexico (ethical approval code, CINVA 271 and 1868) approved all procedures, which are in accordance with the ARRIVE criteria for reporting animal studies [39,40]. Female albino Wistar rats were born and raised in the INCMNSZ animal facility, which is accredited by the Association for Assessment and Accreditation of Laboratory Animal Care International (AAALAC) and follows its standards. The rats were kept at 22–23 °C under controlled lighting (lights on from 07.00 to 19.00 h) with free access to food and water.

### 2.2. Experimental Design

First, 120-day-old female Wistar rats were randomly mated with proven fertile non-littermates to produce the founder generation (F0) of mothers. The F0 litters were adjusted to ten pups at birth (day 0), with at least four females [41]. At weaning (21 days old) F0 females were randomly assigned to one of the two experimental groups: control (F0C) or maternal obesity (F0MO) groups to be fed either a standard laboratory chow diet or a high-fat diet (HFD). The C diet consisted of standard laboratory chow (Zeigler Rodent RQ22-5, Gardners, PA, USA) containing 22.0% protein, 5.0% soy oil fat, 31.0% polysaccharide, 31.0% simple sugars, 4.0% fiber, 6.0% minerals, and 1.0% vitamins (*w*/*w*) (physiological fuel 3.4 kcal/g). The HFD was produced in the INCMNSZ’s specialized dietary facility, with 23.5% protein, 20.0% lard, 5.0% soy oil fat, 20.2% polysaccharide, 20.2% simple sugars, 5.0% fiber, 5.0% mineral mix, 1.0% vitamin mix (*w*/*w*), and physiological fuel 4.8 kcal/g.

At 120 days, 10 female rats from the F0C group and 20 from the F0MO group were mated overnight (up to 5 days) with non-experimental males to produce offspring. Daily vaginal smears were obtained and the day a sperm plug was found was designated as day 0 of conception. To ensure similar pregnancy conditions, this study excluded litters with fewer than 9 or more than 14 pups. In addition, to achieve offspring homogeneity, on the second PND, all offspring litters studied were adjusted to 10 pups, with equal numbers of males and females whenever possible (C and MO). This adjustment to litter size had no effect on the metabolic variables as the litter size was considered normal.

### 2.3. Care and Maintenance of Offspring to Study for Developmental Programming and Aging Interactions

Our studies have routinely been conducted at 110PND, to obtain data at a young adult life stage, and at 650PND, to obtain data at a mature aged adult life stage. The litters were weaned at 21PND and males and females were divided into separate cages at weaning. After weaning, all offspring ate a control diet until the end of the experiment (110 and 650PND). There was no mixing of litters or sexes from different age groups. The offspring were maintained in this situation until PND50, after which, no more than 4 rats were placed in one cage. After 110PND the number was reduced to a minimum of 2 or a maximum of 3 per cage, as previously reported. All females at 110PND were evaluated during the diestrus phase of the ovarian cycle.

### 2.4. Offspring Tissue Collection

One male and one female from different litters were euthanized at 110 and 650PND by exsanguination through aortic punctures under isoflurane general anesthesia; this was conducted by the same experienced person under identical conditions at each timepoint (light period (12:00 to 14:00 h) and 6 h of fasting). Thus, the males and females evaluated at the two ages were groups of siblings. For each age group, the livers were dissected, cleaned, and weighed. The right inferior lobes were fixed in 4% paraformaldehyde and embedded in paraffin for immunohistochemical analysis. The left lobes were stored at −70 °C for RNA-seq analysis. We report data with the following number of animals per group and sex: 110PND—males: C *n* = 6, MO *n* = 5; females: C *n* = 6, MO *n* = 5; 650PND—males: C *n* = 6; MO *n* = 6; females: C *n* = 6 and MO *n* = 6.

### 2.5. RNA Extraction and cDNA Library Preparation and Sequencing

Liver tissue samples (10–20 mg) were homogenized with the BioSpec BeadBeater (BioSpec products, Bartlesville, OK, USA) and RNA was extracted using the Qiagen miRNeasy mini kit (Qiagen, Hilden, Germany), according to the manufacturer’s instructions. RNA quantity and quality were determined using a Nanodrop spectrophotometer (Nanodrop Technologies, Wilmington, DE, USA). RNA was stored at −80 °C until it was used. cDNA libraries were generated from 1 μg of total RNA using an Illumina TruSeq RNA LS Sample Preparation kit v2, according to the manufacturer’s instructions (Illumina, San Diego, CA, USA). The Agilent DNA 1000 was used to evaluate the quality and fragment size of the final individual cDNA libraries. The sequencing libraries were quantified using the KAPA Library Quantification kits for Illumina platforms. The libraries were normalized to 10 nM and diluted to 20 pM before loading on the cBot 2X100; the Illumina HiSeq 2500 sequencer was used for paired-end sequencing.

### 2.6. Bioinformatic Analysis

Output demultiplexed reads were exported to Partek Flow for analysis. Read FASTQ files were trimmed for quality to Phred 30 at each end. STAR aligner v2.3.1j was used to align trimmed reads to the *Rattus norvegicus* genomic reference (RGSC 5.0/rn5). Gene and transcript abundance were quantified against the rn5 RefSeq annotation and transcript abundance was normalized for all samples as a dataset using the Reads Per Kilobase per Million mapped reads (RPKM) values of all Refseq genes. To identify the functional pathways related to maternal obesity programming–aging interactions, we evaluated the differentially expressed genes (DEGs) between 650PND and 110PND in the male and female livers of the C and MO groups using pairwise comparisons with Partek Flow (Partek®, St. Louis, MO, USA). Genes were filtered based on >1 fold change (FC) and a nominal *p*-value of <0.05 (Student’s *t*-test). All DEGs were mapped to the KEGG database (Kyoto Encyclopedia of Genes and Genomes) and searched for principal mitochondrial function-related pathways.

### 2.7. KEGG Pathway Analysis

The Web Gestalt application (WEB-based Gene SeT AnaLysis Toolkit) was used to perform analyses for enrichment KEGG pathways; the statistical significance *p*-value cutoff was set at 0.05 [42]. The KEGG is an online bioinformatics analysis system for over-represented pathways [43].

### 2.8. Liver Immunohistochemical (IHC) Analysis

Each liver’s right inferior lobe was dissected, sectioned longitudinally, and immediately fixed in 4% paraformaldehyde in a neutral phosphate saline buffer. Following a 24 h fixation period, liver sections were dehydrated with ethanol at increasing concentrations from 75 to 95% and were then embedded in paraffin.

IHC analysis was carried out using the avidin–biotin complex (ABC) IHC method. Liver sections (4 μm) were deparaffinized, hydrated, and quenched for endogenous peroxidase with 0.3% hydrogen peroxide in PBS at room temperature for 30 min. To perform antigen retrieval, slides were placed in citrate buffer at pH 6.0 (ImmunoDNA Retriever Citrate, BioSB, Inc., Santa Barbara, CA, USA) and heated in a pressure cooker for 5 min. The sections were then incubated overnight at room temperature. The following primary antibodies were used for IHC analysis: anti Atp5f1, 1:500 (goat polyclonal SC-162552, Santa Cruz Biotechnology, Dallas, TX, USA); anti Ndufa10, 1:500 (mouse monoclonal SC-376357, Santa Cruz); anti Cox5a, 1:100 (mouse monoclonal SC-376907, Santa Cruz Biotechnology); anti Sdhc, 1:1000 (rabbit polyclonal SC-67256, Santa Cruz Biotechnology); anti Sirt-2, 1:1000 (mouse monoclonal SC-28292, Santa Cruz Biotechnology); anti Sirt-3, 1:300 (rabbit polyclonal SC-99143, Santa Cruz Biotechnology); anti Sod-1, 1:1000 (rabbit polyclonal SC-11407, Santa Cruz Biotechnology); Catalase, 1:600 (rabbit polyclonal SC-50508, Santa Cruz Biotechnology). Antibody binding was detected with a Vectastain Elite ABC kit (Vector Laboratories, Burlingame, CA, USA) and 3, 3′-diaminobenzidine as a chromogen. After tissue sections were stained, hematoxylin was used as a counterstain. Negative controls were performed without the primary antibody. Due to space limitations, the negative staining controls are presented as Appendix A. Twenty random digital images were taken of each rat using an Olympus BX51 microscope (Olympus Co. Model BX51RF, Tokyo, Japan). The staining areas of the images were analyzed using digital image analyzing software (ImageJ, U.S. National Institute of Health, Bethesda, MD, USA) and a color deconvolution plug-in.

### 2.9. Statistical Analysis

Gene expression is expressed as mean Log2 RPKM± standard error of the mean (SEM). Immunohistochemical analyses are presented as mean ± SEM. A *p*-value < 0.05 was considered statistically significant. To analyze differences between the groups, we used the Tukey test (one-way ANOVA) for males and females separately. Analysis was performed with the Sigma Stat 3.5 statistical program (2005). Gene expression from RNA-seq data is shown as the mean of Log2FC and SEM based on normalized data. There was no overlap in DEGs between the sexes in the pathways examined, indicating that it was not necessary to compare males and females to determine sexual dimorphism.

## 3. Results

### 3.1. Liver Differentially Expressed Genes (DEGs)

We performed four different comparisons to evaluate the effects of maternal obesity at two different ages (young and old) and the effect of aging in the control group’s and obese mothers group’s offspring. The comparisons were as follows: (1) the effect of maternal diet on the young (MO-110PND vs. C-110PND); (2) the effect of maternal diet on the old (MO-650PND vs. C-650PND); (3) the effect of aging on the control groups (C-650PND vs. C-110PND); and 4) the effect of maternal obesity on aging (MO-650PND vs. MO-110PND). Males and females were analyzed separately.

The number of DEGs for male comparisons: (1) MO-110PND vs. C-110PND showed that 3030 genes were down- and 118 genes were up-regulated; (2) MO-650PND vs. C-650PND revealed that 35 genes were down- and 604 were up-regulated; (3) C-650PND vs. C-110PND indicated that 4218 genes were down- and 127 were up-regulated; (4) MO-650PND vs. MO-110PND showed that 480 genes were down- and 1285 were up-regulated.

The number of DEGs for female comparisons: (1) MO-110PND vs. C-110PND exhibited that 51 genes were down- and 127 genes were up-regulated; (2) MO-650PND vs. C-650PND revealed 9244 down- and 3 up-regulated genes; (3) C-650PND vs. C-110PND showed that 57 genes were down- and 415 were up-regulated; (4) MO-650PND vs. MO-110PND exhibited 3346 down- and 44 up-regulated genes.

Based on the DEG analysis for the effects of maternal obesity, we observed that 96% of the genes were down-regulated in male comparisons in both maternal obesity (MO-110PND vs. C-110PND) and age (C-650PND vs. C-110PND). We performed a Venn analysis to determine if the DEGs in both conditions shared common genes; then, we evaluated whether these genes are associated with premature aging (Figure 1).

The Venn diagram shows the distribution of DEGs by maternal diet in the young and by age in the control groups for males (Figure 1A) and for females (Figure 1B). Male Venn diagram with DEG comparisons: C-650PND vs. C-110PND and MO-110PND vs. C-110PND reveal that there are 1477 genes in common for both comparisons; we refer to these genes as genes involved in premature aging due to MO. In the same comparison, females only shared 35 genes. Clearly, maternal diet affects more genes involved in premature aging in males compared to females.

### 3.2. KEGG Pathway Analysis for Prematurely Aging Genes in Males and Females

Using the lists of 1477 and 35 common genes (premature aging) in males and females, respectively, we performed an over-represented analysis to identify the KEGG pathways enriched from these two DEG lists (Appendix A). The two lists were mapped to KEGG pathways separately; we found that fifty-one KEGG pathways showed significant changes in the male livers but just three did so in the livers of the females. Table 1 shows the most significant KEGG pathways (by *p*-value). In males (Table 1A), the pathways related to liver metabolism and aging, oxidative phosphorylation, and NAFLD are at the top of the KEGG pathway analysis; however, in females (Table 1B), there were only three pathways that changed significantly.

### 3.3. Pathway Analysis Related to Mitochondria

To determine that the pathways implicated in the MO-110PND vs. C-110PND and C-650PND vs. C-110PND comparisons were enriched and significant separately, we evaluated the over-represented analysis of the pathways of DEGs in each comparison using three different databases: KEGG, Wikipathway, and Reactome. In Table 2A–C, the results of the male DEGs in each database are displayed. For the analysis, we focused on mitochondrial function-related pathways; for each comparison, oxidative phosphorylation was significantly enriched and all DEGs in the pathways were down-regulated.

### 3.4. Oxidative Phosphorylation KEGG Pathway

In accordance with the aims of this study, we restricted our detailed analysis to changes in genes related to the oxidative phosphorylation KEGG pathway (Table 3). This pathway showed no overlap in DEGs between sexes or diet comparisons, clearly demonstrating a sex-dependent aging and maternal diet effect on the liver transcriptome.

Figure 2 shows the overlapping genes between the comparisons for aging (1) C-650PND vs. C-110PND and for programming by obesity (2) MO-110PND vs. C-110PND. Genes that were in common in these two comparisons were involved in premature liver aging, specifically in the oxidative phosphorylation KEGG pathway (Figure 2A) and on the ETC Wikipathway (Figure 2B).

### 3.5. Male and Female Liver Oxidative Phosphorylation Complexes

We selected one representative gene from each oxidative phosphorylation complex. In males, the genes *ndufa10* (Complex I), *sdhc* (Complex II), *cox5a* (Complex IV), and *atp5f1* (Complex V) were down-regulated in the groups MO-110PND, C-650PND, and MO-650PND in comparison to the C-110PND group (Figure 3A–D). In contrast, the gene expression of *ndufa10*, *sdhc*, *cox5a*, and *atp5f1* was down-regulated only in MO-650PND females (Figure 4A–D).

To determine whether changes in gene expression are associated with changes in gene protein products, proteins encoded by the *ndufa10*, *sdhc*, *cox5a, and atp5f1* genes were quantified by an IHC analysis. Males in the MO-110PND, C-650PND, and MO-650PND groups exhibited a lower liver fractional area being stained for Sdhc and Atp5f1 proteins than those in the C110-PND group; for the Ndufa10 and Cox5b proteins, only the MO-650PND group differed from the C-110PND group (Figure 3E–H). In contrast, the results for females varied across all proteins evaluated. The percentage of the Ndufa10 protein’s stained area was higher in MO-650PND compared to C-110PND, with no differences between the MO-110PND and C-650PND groups. In terms of the Sdhc protein, C650-PND and MO-650PND had higher percentages of stained areas compared to the C and MO groups at younger ages. The Cox5a protein was similar in all groups; whereas, the groups from obese mothers, regardless of age, exhibited lower percentages of stained areas for the Atp5f1 protein (Figure 4E–H). Figure 3I and Figure 4I show representative sections stained by IHC analysis for all of the oxidative phosphorylation proteins for males and females, respectively.

### 3.6. Male and Female Liver Sirtuins

The *sirt-2* mRNA expression and protein content were both decreased in all groups in comparison to the C-110PND group (Figure 5A,B). Despite C-650PND exhibiting the lowest level of *sirt-3* gene expression, the protein abundance did not differ from C110PND; for the groups representing maternal obesity (MO-110PND and MO-650PND), both gene and protein contents were lower compared to C-110PND (Figure 5D,E). Figure 5C,F show representative sections stained by IHC analysis for the Sirt-2 and Sirt-3 proteins, respectively.

In terms of sirtuin gene expression in female livers, only the MO-650PND group showed a decrease in *sirt-2* and *sirt-3* (Figure 6A and D, respectively). However, unexpectedly Sirt-2 and Sirt-3 protein content was higher in C-650PND, and even higher in MO-650PND, compared to MO-110PND and C-110PND (Figure 6B,E). Figure 6C,F show the representative sections stained by IHC analysis for the Sirt-2 and Sirt-3 proteins, respectively.

### 3.7. Male and Female Liver Sod-1 and Catalase

For males, the groups C-650PND and MO-650PND presented less gene expression and a lower protein percentage of the area stained for Sod-1 and Cat compared to C-110PND (Figure 7A,B,D,E). Figure 7C,F show representative sections stained by IHC analysis for *Sod-1* and *catalase* proteins, respectively.

In females, *sod-1* and *catalase* gene expression were only down-regulated in the MO-650PND group in comparison to all groups (Figure 8A and D, respectively). However, when it came to the Sod-1 protein, MO-650PND had a higher protein percentage of area stained than C-110PND and C-650PND (Figure 8B). Regarding the catalase protein, C-650PND and MO-650PND exhibited higher protein concentrations than C-110PND and MO-110PND (Figure 8E). Figure 8C,F show representative sections stained by IHC analysis for the Sod-1 and catalase proteins, respectively.

## 4. Discussion

Exposure to a high-fat diet prior to and/or during pregnancy and lactation has long-term consequences for both mothers and their offspring. Maternal obesity increases offspring liver fat accumulation, which negatively affects offspring metabolism and predisposes neonates and children to obesity and NAFLD, increasing oxidative damage, inflammation, insulin resistance, lipid metabolism, and mitochondrial function [31]. The fetal liver function is immature and vulnerable to dysregulated maternal metabolism. Exposing the fetus to an excess of metabolic fuels from an obese mother during gestation contributes to the programming of NAFLD in childhood [36,38,44]. Studies in rodents [4,10,45], ewes [15], and non-human primates [46] have shown that maternal obesity programs offspring to develop hepatic metabolic disorders later in life and correlates with the severity of childhood NAFLD [47].

There is also considerable interest in the potential that developmental programming can alter the trajectory of aging [48,49,50]. Aging is a multifactorial degenerative process in which physiological and metabolic processes decline and is a risk factor for the development of metabolic diseases [51]. The natural biological changes that occur during aging differ among major organs and are sexually dimorphic [52]. In this regard, males aged earlier than females [53]. In humans, fatty liver disease is more severe and has a worse prognosis in the elderly than in young adults [54].

In addition to alterations in genes, proteins, and metabolites, liver aging is accompanied by a redox imbalance and a decline in hepatic metabolism. Among the alterations associated with liver aging, several signaling pathways are implicated, such as those related to xenobiotic metabolism, lipid metabolism, oxidative stress [55], cell growth [56], immune cell responses [53,57], metabolic processes, cell activation [57], and inflammatory processes [58,59].

In our animal model, the male and female offspring of obese mothers have higher adiposity indexes, triglycerides, and insulin resistance compared to those of control mothers. Males from the MO group exhibit greater physiological and histological NAFLD characteristics than females at 110 days [4]. Also, human studies indicate that the prevalence of NAFLD is higher in men [60]. However, little is known about developmental programming–aging interactions and the molecular mechanisms of NAFLD programmed by MO.

Our observations showed that where genes that were down-regulated in C-650PND vs. C-110PND (natural aging) were also down-regulated in the MO-110PND vs. C-110PND comparison (effect of maternal diet at a young age) they were considered to be genes programmed by maternal obesity to age prematurely. Furthermore, at a young age, we observed sex differences in the gene expression profiles between the offspring of obese mothers and offspring born to the control mothers, as well as in the natural liver aging course in our animal model at old age (650PND) vs. young age (110PND), with males again having more pronounced changes. Clearly, in all studies, the sexual dimorphism of the outcomes must be addressed in determining the underlying mechanisms involved.

In the MO-programmed NAFLD phenotype, we have also previously observed important metabolic and liver oxidative stress sexual dimorphism. In the liver transcriptomic analysis, we observed diet and age effects in a sex-dependent manner regarding mitochondrial pathways. In our model, we reported the phenotypic characterization of NAFLD in the offspring of obese mothers [10]. The changes observed in MO offspring compared to C offspring were programmed by the mother’s consumption of a high-fat diet; since offspring were weaned onto a C diet and did not consume a high-fat diet, the observed changes in the NAFLD phenotype in gene expression and protein concentration were programmed from fetal and neonatal exposure to excess fetal nutrients. Importantly, changes in mitochondrial function have been demonstrated from fetal and neonatal ages, prior to the establishment of NAFLD [31].

The liver is an organ that plays a central role in the body’s main metabolic processes, including energy production, and is therefore essential for regulating energy balance [61]. In this regard, oxidative phosphorylation is by far the principal pathway for cellular energy production and is the primary source of ROS production [62]. Aging induces morphological, structural, and functional changes in the liver, as well as increased levels of ROS, oxidative damage, decreased mitochondrial energy production capacity, and dysfunction of the respiratory chain [62,63,64]. Among the molecular mechanisms of NAFLD programmed by MO, major pathways and genes related to mitochondrial function, such as lysosome, ribosome, peroxisome, TCA cycle, mitophagy, ETC, oxidative stress, and oxidative phosphorylation, are involved in premature aging in males. Therefore, mitochondria and peroxisomes are significant ROS sources [64].

We studied genes involved in oxidative phosphorylation as possible contributors to increased ROS concentrations. During both normal aging and accelerated aging in the offspring of obese mothers, a number of sex-related gene-expression changes were detected. Age-related declines in mitochondrial function and antioxidant enzymes result in a rise in mitochondrial ROS production. Different studies comparing old and young animals have evaluated mitochondrial function and found that the number of mitochondria and the mitochondrial protein concentrations decrease with age in the liver cells of mice, rats, and humans [24]. In addition, the respiratory chain capacity of liver mitochondria in aged rats (720 days) is reduced by 40% compared to young rats (90–120 days) [25]. Mitochondrial dysfunction is one of the hallmarks of aging [22,23,24,25,26,27,65,66] and is related to the progression of NAFLD.

In males, maternal obesity and aging led to the down-regulation of representative genes for Complexes I, II, IV, and V. These findings are consistent with the decreased immunolocalization of Complexes I and IV, mainly in the MO groups, as well as the decreased immunolocalization of Complexes II and V observed in the MO and aged groups. In fact, the activity of Complexes I and IV decreased with age in the livers of mice and rats; whereas, the activity of Complexes II, III, and IV remained relatively unchanged [25]. In a mouse model of maternal obesity, it has been reported that 105-day-old female offspring reduced the hepatic mitochondrial ETC activity of Complexes I, II/III, and IV [32]. This observation suggests the presence of post-translational mechanisms in ETC-associated gene expression. Maternal obesity also programs increased adiposity in males and females [7], which worsens with age [10]. In this regard, it is known that obesity alone accelerates aging and adipose tissue dysfunction can be observed earlier than in normal aging [67]. In addition, the continued delivery of FFAs to liver mitochondria induces a hypermetabolic state, as occurs with insulin resistance, which further impairs mitochondrial bioenergetics in the adipocytes of diabetic (*db/db*) individuals. This situation may resemble our model in which the offspring of obese mothers accumulated much dysfunctional adipose tissue with signs of insulin resistance in which the suppressed expression of mitochondrial proteins caused mitochondrial loss, decreased fatty acid oxidation, and lowered ATP production [68]. Thus, the loss of mitochondrial function plays an important role in the progression of NAFLD [29].

MO led to the down-regulation of representative genes for Complexes I, II, IV, and V, only in females at 650PND. Nevertheless, the increased immunolocalization of Complexes I and II was mainly observed in aged offspring MO females. These findings suggest an adaptation mechanism for offspring MO females as they age. Compared to males, females have greater respiratory function and mitochondrial biogenesis in several tissues [69]. In addition, females exhibit a tighter regulation of mitochondrial processes than males, which affords them increased protection in the presence of metabolic challenges [70]. In this regard, the ETC can be regulated through the expression of Complexes I and II by tuning the availability of NADH and succinate [71]. In addition, the observed increased immunostaining for Complex II, which catalyzes the oxidation of succinate to fumarate, suggests a mechanism for protecting the integrity of the TCA cycle and oxidative phosphorylation [72]. Regardless of age, the decreased immunolocalization of Complex V was observed mainly in the MO groups for both sexes. The primary function of mitochondria is to generate ATP via oxidative phosphorylation, which is carried out by the four respiratory chain complexes (I–IV) and the ATP synthase (Complex V), which are localized within the mitochondrial inner membrane.

Through the ATP synthase system, the ETC is tightly coupled with the oxidative phosphorylation pathway to enable the production of metabolically useful energy in the form of ATP. The lack of consistency observed between the transcription and expression of mitochondrial complexes in each sex indicates that mitochondrial biogenesis is sex-dependent. The increased expression of *sirt-2* (from the cytoplasm) and *sirt-3* (from the mitochondria) that we observed in females in the offspring of the MO 650PND group may be related to a potential increase in mitochondrial biogenesis as SIRTs are known to indirectly regulate the expression of mitochondrial biogenesis through PGC-1 activation.

NAFLD is graded as simple steatosis, nonalcoholic steatohepatitis (NASH), liver cirrhosis, or hepatocellular cancer [73]. The progression from simple steatosis to NASH involves the generation of reactive oxygen species, lipotoxicity, and inflammatory cytokines [74]. The sirtuins family are highly conserved NAD^+^-dependent histone deacetylases that have been related to antioxidant and oxidative stress-related processes and functions like longevity, mitochondrial function, DNA-damage repair, and metabolism [75]. In mammals, seven members (*sirt1-7*) have been identified, with *sirt-2* being the least recognized but highly expressed in metabolically active tissues, including the liver, heart, brain, and adipose tissue [76]. In obese mice, it has been shown that *sirt-2* hepatic overexpression ameliorates insulin sensitivity, oxidative stress, and mitochondrial dysfunction [77]. However, a link between *sirt-2* and NAFLD has not yet been established. In obese *ob/ob* mice and HFD-fed mice, it has been reported that liver Sirt-2 protein levels gradually decreased with age. This reduction was also confirmed in HepG2 cells treated with palmitate in a time- and dose-dependent manner, indicating that hepatic *sirt-2* expression significantly decreased in the context of NAFLD [78]. Our findings showed that maternal obesity decreased liver *sirt-2* expression in offspring in a sex-dependent manner, with the reduction starting at PND110 in MO males and PND650 in females. The *sirt-3* is also highly expressed in the liver and other metabolic tissues with high oxidative capacity. Sirt-3 plays an important role in mitochondrial metabolism through the reversible acetylation of mitochondrial proteins [79]. Low Sirt-3 activity, mitochondrial dysfunction, and protein hyperacetylation were observed in the liver of mice fed a chronic HFD [80]. In a separate study, *sirt-3*-deficient mice fed a chronic HFD developed obesity, insulin resistance, and steatohepatitis more rapidly than wild-type mice [81]. Our findings showed that maternal obesity decreased liver *sirt-3* expression in a sex-dependent manner, with the reduction starting at 110PND in MO males and 650PND in MO females. Therefore, the reduction in *sirt-2* and *sirt-3* expression might be related to the decline in the expression of antioxidant enzyme genes and the increased reactive oxygen species, oxidative stress, and fatty liver accumulation [10]. It is well known that oxidative stress contributes to aging. During the aging process, cells defend themselves against oxidative damage by expressing a variety of non-enzymatic and enzymatic antioxidant defenses that convert ROS into less dangerous byproducts. Sod converts the anion superoxide to hydrogen peroxide and it mitigates the ROS produced by the mitochondria; but, as NAFLD progresses, Sod decreases. In the present study, age and diet reduced *sod* gene expression in both males and females. However, the amount of protein was higher in MO-650PND compared to MO-110PND and the C group. This observation merits further study. It may be due to post-translational changes in protein production. In mice, the deletion of liver *sod-1* accelerates aging, shortens the life span, and results in the development of hepatocellular carcinoma [82].

The observed changes in gene and protein expression associated with the mitochondrial function pathways (Figure 9), together with those previously observed in our experimental model, such as insulin resistance, increased liver fat accumulation, visceral fat and oxidative stress, decreased antioxidant enzymes, and liver morphological alterations in MO offspring, contribute to the programming of the MO offspring fatty liver phenotype. Likewise, the differences in the changes in gene expression observed between the diet, age, and sex comparisons could be associated with the severity of the fatty liver over the life span as the changes observed at 110PND remain at age 650PND. This study links mitochondrial function to signaling pathways that regulate the life span and the aging process; it demonstrates the role and importance of the mitochondria in the predisposition to developing NAFLD.

## 5. Conclusions

Maternal obesity programs sex-specific changes associated with the natural aging process leading to liver dysfunction in offspring. In males at 110PND, maternal obesity accelerates the age-associated down-regulation of genes and pathways related to mitochondrial function. In females, these programming effects occur at 650PND. Moreover, maternal obesity programs decreased offspring liver ETC gene expression, especially Complex 1, the major site of ROS production. These changes can lead to metabolic dysfunction and offspring obesity and are potential mechanisms for programming offspring from maternal obesity life-course metabolic dysfunction.

## Figures and Tables

**Figure 1 biology-12-01166-f001:**
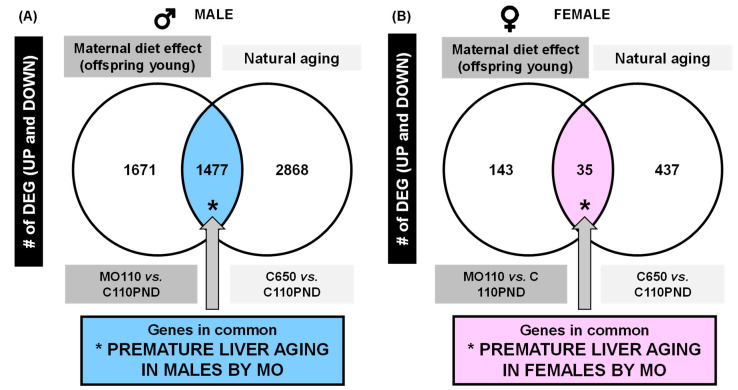
Venn diagram of up- and down-regulated liver genes, (**A**) males and (**B**) females.

**Figure 2 biology-12-01166-f002:**
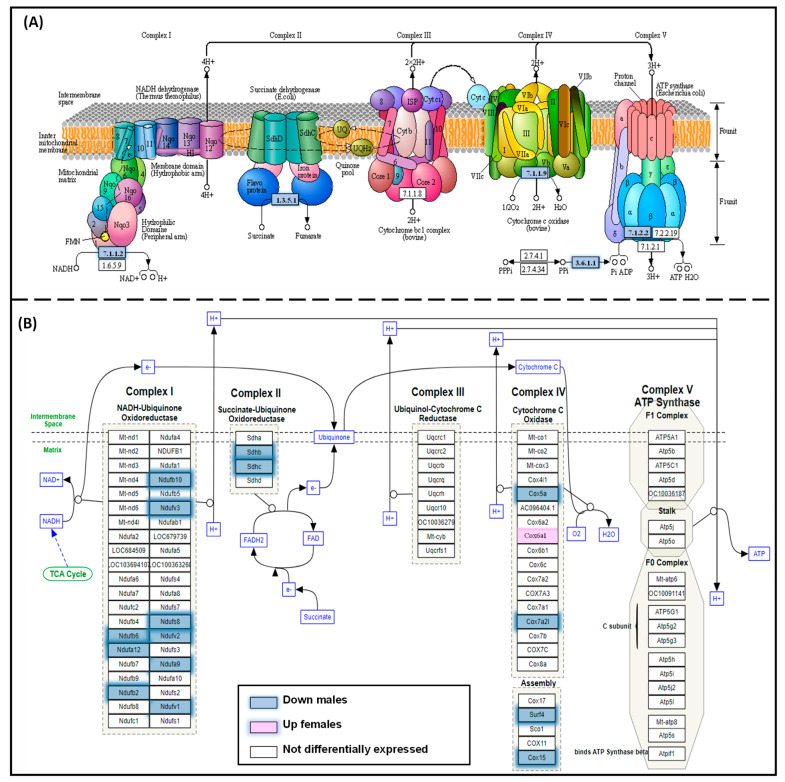
Overlapping genes for (**A**) the oxidative phosphorylation KEGG pathway and (**B**) the electron transport chain (Wikipathway) between the comparisons C-650PND vs. C-110PND and MO-110PND vs. C-110PND, enriched with common DEGs of male premature aging. Genes that were down-regulated in both comparisons are indicated in blue.

**Figure 3 biology-12-01166-f003:**
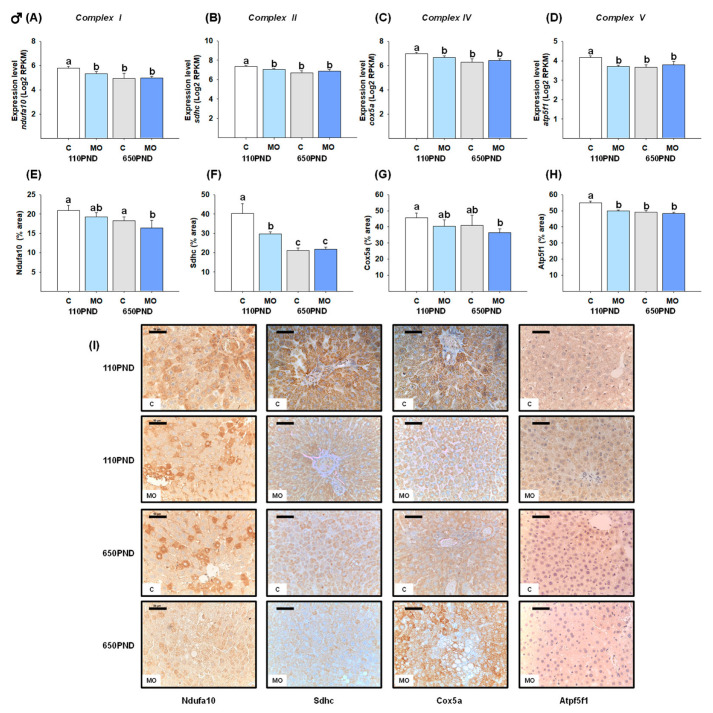
Male hepatic gene expression and protein abundance of four proteins in oxidative phosphorylation complexes in the control (C) and maternal obesity (MO) groups. Gene expression level (Log2 RPKM) of (**A**) *ndufa10;* (**B**) *sdhc*; (**C**) *cox5a*; (**D**) *atp5f1*; the immunostained area (%) of (**E**) Ndufa10; (**F**) Sdhc; (**G**) Cox5a; (**H**) Atp5f1; and (**I**) representative IHC micrograph (40×). Data for RNA-seq, mean Log2 RPKM ± SEM. Protein values are mean ± SEM. *p* < 0.05 for data not sharing a lowe case letter between groups. *N* = 5–6 rats/group/litter. PND = Postnatal days. Scale bar: 50 μm.

**Figure 4 biology-12-01166-f004:**
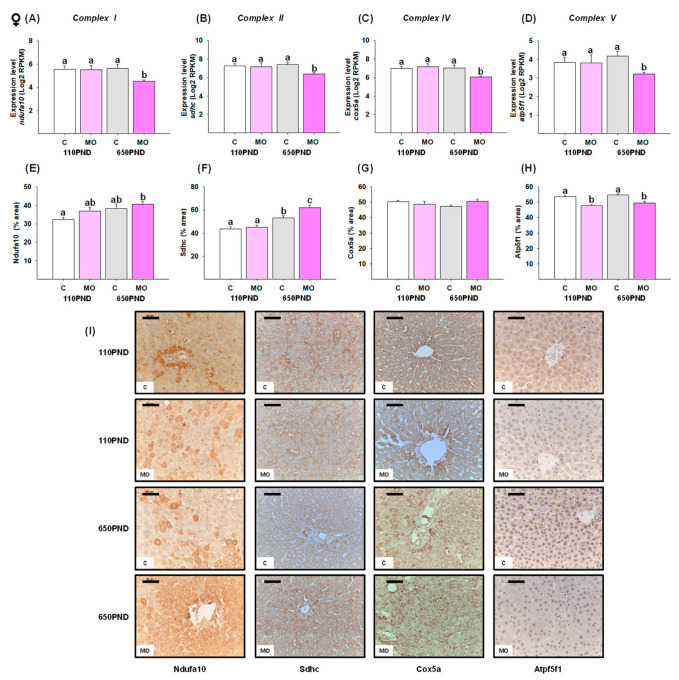
Female hepatic changes in gene expression and abundance of proteins in four oxidative phosphorylation complexes in the control (C) and maternal obesity (MO) groups. Gene expression (Log2 RPKM) of (**A**) *ndufa10;* (**B**) *sdhc*; (**C**) *cox5a*; (**D**) *atp5f1*; the immunostained area (%) of (**E**) Ndufa10; (**F**) Sdhc; (**G**) Cox5a; (**H**) Atp5f1; and (**I**) representative IHC micrograph (40×). Data for RNA-seq, mean Log2 RPKM ± SEM. Protein values are mean ± SEM. *p* < 0.05 for data not sharing a lower case letter between groups. *n* = 5–6 rats/group/litter. PND = Postnatal days. Scale bar: 50 μm.

**Figure 5 biology-12-01166-f005:**
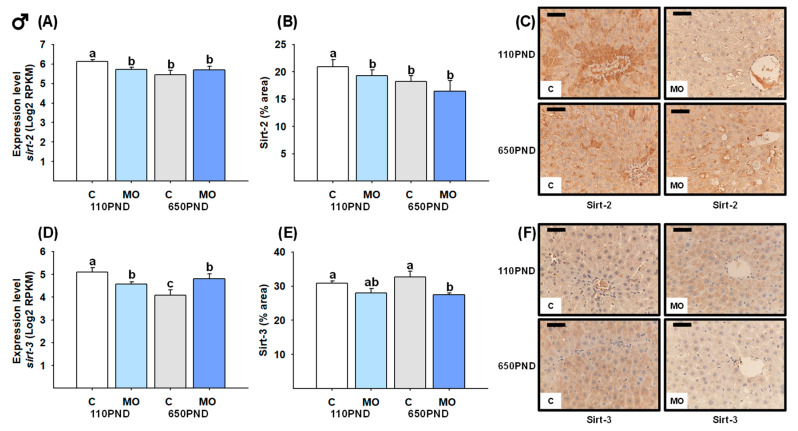
Male hepatic gene expression and protein abundance of Sirt-2 and Sirt-3 in the control (C) and maternal obesity (MO) groups. (**A**) Expression level of *sirt-2* (Log2 RPKM); (**B**) Sirt-2 immunostained area (%); (**C**) representative IHC micrograph of Sirt-2 (40×); (**D**) expression level of *sirt-3* (Log2 RPKM); (**E**) Sirt-3 immunostained area (%); and (**F**) representative IHC micrograph of Sirt-3 (40×). Data for RNA-seq, mean Log2 RPKM ± SEM. Protein values are mean ± SEM. *p* < 0.05 for data not sharing a lower case letter between groups. *n* = 5–6 rats/group/litter. PND = Postnatal days. Scale bar: 50 μm.

**Figure 6 biology-12-01166-f006:**
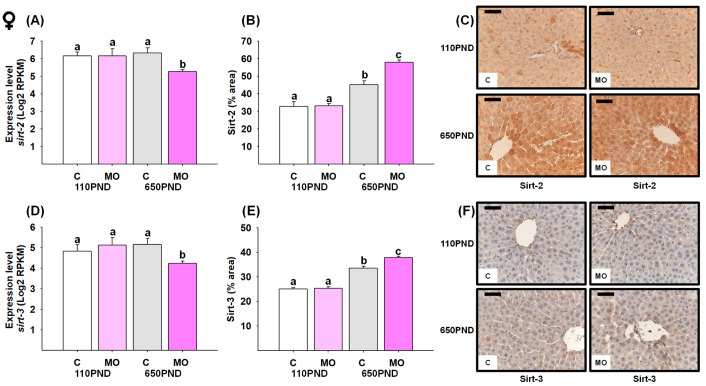
Female hepatic gene expression and protein abundance of Sirt-2 and Sirt-3 in the control (C) and maternal obesity (MO) groups. (**A**) Expression level of *sirt-2* (Log2 RPKM); (**B**) Sirt-2 immunostained area (%); (**C**) representative IHC micrograph of Sirt-2 (40×); (**D**) expression level of *sirt-3* (Log2 RPKM); (**E**) Sirt-3 immunostained area (%); and (**F**) representative IHC micrograph of Sirt-3 (40×). Data for RNA-seq, mean Log2 RPKM ± SEM. Protein values are mean ± SEM. *p* < 0.05 for data not sharing a lower case letter between groups. *n* = 5–6 rats/group/litter. PND = Postnatal days. Scale bar: 50 μm.

**Figure 7 biology-12-01166-f007:**
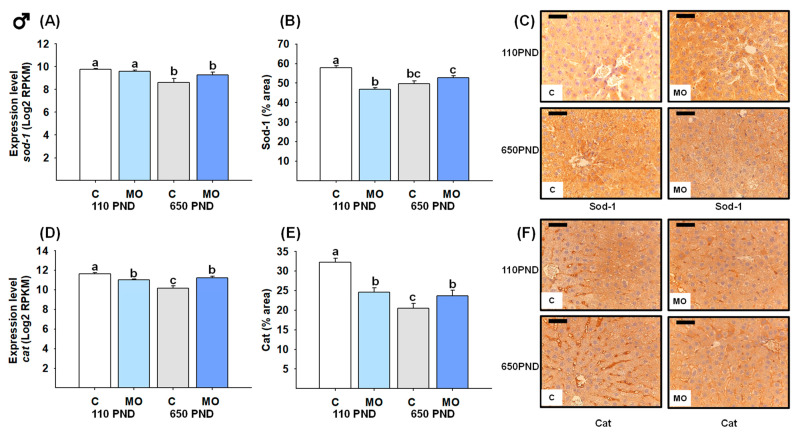
Male hepatic gene expression and protein abundance of Sod-1 and Catalase in the control (C) and maternal obesity (MO) groups. (**A**) Expression level of *sod-1* (Log2 RPKM); (**B**) Sod-1 immunostained area (%); (**C**) representative IHC micrograph of Sod-1 (40×); (**D**) expression level of *catalase* (Log2 RPKM); (**E**) Cat immunostained area (%); and (**F**) representative IHC micrograph of Cat (40×). Data for RNA-seq, mean Log2 RPKM ± SEM. Protein values are mean ± SEM. *p* < 0.05 for data not sharing a lower case letter between groups. *n* = 5–6 rats/group/litter. PND = Postnatal days. Scale bar: 50 μm.

**Figure 8 biology-12-01166-f008:**
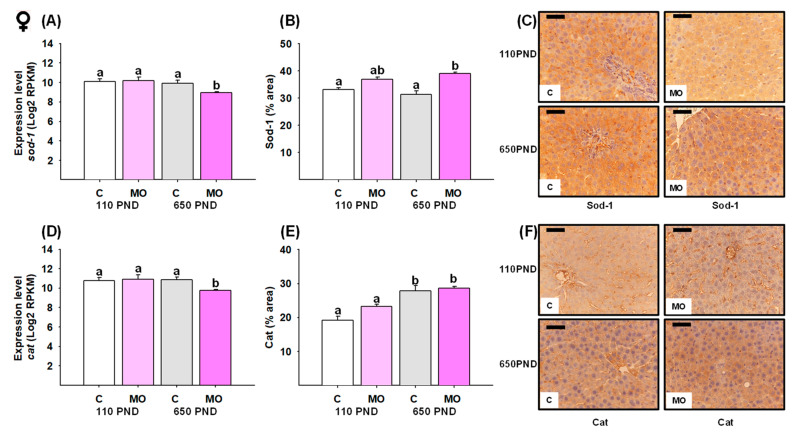
Female hepatic gene expression and protein abundance of Sod-1 and catalase in the control (C) and maternal obesity (MO) groups. (**A**) Expression level of *sod-1* (Log2 RPKM); (**B**) Sod-1 immunostained area (%); (**C**) representative micrograph of Sod-1 (40×); (**D**) expression level of *cat* (Log2 RPKM); (**E**) Cat immunostained area (%); and (**F**) representative micrograph of Cat (40×). Data for RNA-seq, mean Log2 RPKM ± SEM. Protein values are mean ± SEM. *p* < 0.05 for data not sharing a lower case letter between groups. *n* = 5–6 rats/group/litter. PND = Postnatal days. Scale bar: 50 μm.

**Figure 9 biology-12-01166-f009:**
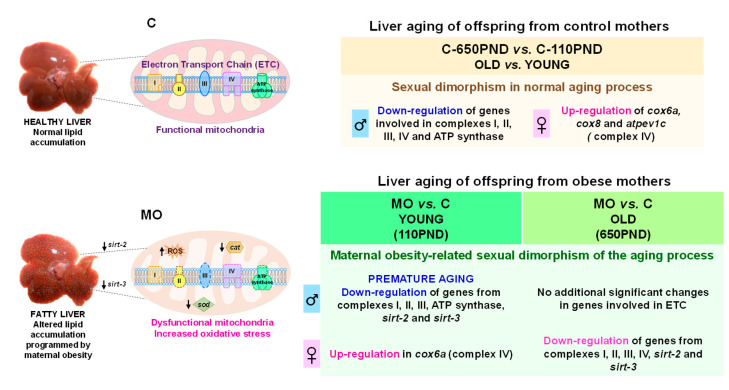
Summary of findings.

**Table 1 biology-12-01166-t001:** List of the most significant KEGG pathways enriched with DEGs by *p*-value (up- and down-regulated), showing premature aging for males (**A**) and females (**B**).

(A). MALE.			
Id Pathway	Name	Size	*p*-Value
rno01100	Metabolic pathways	1380	<2.2 × 10^−16^
rno00280	Valine, leucine, and isoleucine degradation	56	1.7 × 10^−6^
rno00640	Propanoate metabolism	32	1.0 × 10^−5^
rno00190	Oxidative phosphorylation	143	4.1 × 10^−5^
rno04932	Non-alcoholic fatty liver disease (NAFLD)	159	9.4 × 10^−5^
rno04714	Thermogenesis	243	9.5 × 10^−5^
rno00260	Glycine, serine, and threonine metabolism	40	1.0 × 10^−4^
rno00310	Lysine degradation	61	1.1 × 10^−4^
rno00140	Steroid hormone biosynthesis	84	1.1 × 10^−4^
rno01200	Carbon metabolism	127	1.4 × 10^−4^
rno00270	Cysteine and methionine metabolism	49	1.7 × 10^−4^
rno00630	Glyoxylate and dicarboxylate metabolism	30	2.2 × 10^−4^
rno03022	Basal transcription factors	45	1.3 × 10^−3^
rno00760	Nicotinate and nicotinamide metabolism	32	1.8 × 10^−3^
rno00380	Tryptophan metabolism	47	1.9 × 10^−3^
rno04122	Sulfur relay system	9	2.8 × 10^−3^
rno04142	Lysosome	129	2.8 × 10^−3^
rno00510	*N*-Glycan biosynthesis	51	3.6 × 10^−3^
rno03420	Nucleotide excision repair	47	6.6 × 10^−3^
rno04144	Endocytosis	275	6.6 × 10^−3^
rno04217	Necroptosis	161	7.5 × 10^−3^
rno04120	Ubiquitin mediated proteolysis	141	7.6 × 10^−3^
rno00670	One carbon pool by folate	18	8.3 × 10^−3^
rno00830	Retinol metabolism	85	9.1 × 10^−3^
rno03060	Protein export	26	1.0 × 10^−2^
rno00053	Ascorbate and aldarate metabolism	27	1.2 × 10^−2^
rno00563	Glycosylphosphatidylinositol (GPI)-anchor biosynthesis	27	1.2 × 10^−2^
rno00650	Butanoate metabolism	28	1.5 × 10^−2^
rno04141	Protein processing in the endoplasmic reticulum	164	1.8 × 10^−2^
rno00071	Fatty acid degradation	47	2.0 × 10^−2^
rno00350	Tyrosine metabolism	40	2.5 × 10^−2^
rno04146	Peroxisome	88	2.8 × 10^−2^
rno00410	Beta-Alanine metabolism	33	3.2 × 10^−2^
rno00730	Thiamine metabolism	17	3.2 × 10^−2^
rno00920	Sulfur metabolism	10	3.3 × 10^−2^
rno00330	Arginine and proline metabolism	52	3.5 × 10^−2^
rno00010	Glycolysis/Gluconeogenesis	72	3.8 × 10^−2^
rno00980	Metabolism of xenobiotics by cytochrome P450	74	4.5 × 10^−2^
rno03040	Spliceosome	138	4.7 × 10^−2^
**(B). FEMALE**			
**Id pathway**	**Name**	**Size**	***p*-Value**
rno04064	NF-kappa B signaling pathway	97	5.4 × 10^−3^
rno00230	Purine metabolism	182	1.8 × 10^−2^
rno04060	Cytokine–cytokine receptor interaction	269	3.8 × 10^−2^

**Table 2 biology-12-01166-t002:** List of (**A**) KEGG, (**B**) Wikipathway, and (**C**) Reactome enrichment pathways from DEGs (up- and down-regulated) in male livers from the MO-110PND vs. C-110PND and C-650PND vs. C-110PND comparisons.

(A). KEGG Pathway				
Comparison		*p*-Value	Genes Down	Genes Up
MO-110PND vs. C-110PND	Oxidative phosphorylation	7.8 × 10^−5^	39	0
Maternal diet effect (young)	Lysosome	1.7 × 10^−4^	37	0
	Ribosome	3.0 × 10^−3^	41	0
	Peroxisome	9.5 × 10^−3^	23	0
	Citrate cycle (TCA cycle)	1.1 × 10^−2^	11	0
C-650PND vs. C-110PND	Peroxisome	<2.2 × 10^−16^	55	0
Aging effect in controls	Oxidative phosphorylation	2.9 × 10^−14^	68	0
	Mitophagy	1.6 × 10^−4^	27	0
	Lysosome	8.6 × 10^−4^	43	0
	Citrate cycle (TCA cycle)	7.2 × 10^−2^	11	0
**(B). Wikipathway**				
**Comparison**		***p*-Value**	**Genes Down**	**Genes Up**
MO-110PND vs. C-110PND	Oxidative phosphorylation	1.0 × 10^−3^	22	0
Maternal diet effect (young)	Electron Transport Chain	1.5 × 10^−3^	30	0
	TCA Cycle	1.7 × 10^−2^	10	0
	Oxidative Stress	1.9 × 10^−2^	12	0
C-650PND vs. C-110PND	Electron Transport Chain	1.0 × 10^−10^	51	0
Aging effect in controls	Mitochondrial LC-Fatty Acid Beta-Oxidation	1.5 × 10^−7^	14	0
	Oxidative phosphorylation	8.1 × 10^−7^	32	0
	Oxidative stress	1.4 × 10^−5^	20	0
	TCA Cycle	7.3 × 10^−2^	10	0
**(C). Reactome**				
**Comparison**		***p*-Value**	**Genes Down**	**Genes Up**
MO-110PND vs. C-110PND	Mitochondrial translation termination	5.2 × 10^−14^	46	0
Maternal diet effect (young)	Mitochondrial translation	9.5 × 10^−14^	46	0
	The citric acid (TCA) cycle and respiratory electron transport	1.3 × 10^−6^	46	0
	Respiratory electron transport	7.1 × 10^−5^	25	0
	Citric acid cycle (TCA cycle)	9.8 × 10^−4^	10	0
	Pyruvate metabolism and Citric Acid (TCA) cycle	1.3 × 10^−3^	17	0
	Peroxisomal protein import	1.9 × 10^−3^	18	0
C-650PND vs. C-110PND	Mitochondrial translation	<2.2 × 10^−16^	66	0
Aging effect in controls	Mitochondrial translation termination	<2.2 × 10^−16^	66	0
	Mitochondrial translation elongation	<2.2 × 10^−16^	65	0
	Peroxisomal protein import	2.5 × 10^−10^	47	0
	Respiratory electron transport	3.4 × 10^−10^	39	0

**Table 3 biology-12-01166-t003:** List of DEGs for males and females in each comparison of the oxidative phosphorylation KEGG pathway.

Comparison	Genes, Male	*p*-Value	Genes, Female	*p*-Value
(1) MO-110PND vs. C-110PND Maternal diet effect (young)	*atp5d, atp5g2, atp5i, atp5o, atp6v0a1, atp6v1f, cox15, cox5b, cox7a2l, cyc1, lhpp, ndufa10l1, ndufa11, ndufa12, ndufa9, ndufb10, ndufb11, ndufb2, ndufb3, ndufb6, ndufb8, ndufc2, ndufs1, ndufs2, ndufs7, ndufs8, ndufv1, ndufv2, ndufv3, ppa2, sdha, sdhb, tcirg1, uqcr11, uqcrc1, uqcrc2,*	7.8 × 10^−5^	** *cox6a* **	---
(2) MO-650PND vs. C-650PND Maternal diet effect (old)	*atp6v0a4, **atp6v0a2***	----	*atp5f1a, atp5f1b, atp5f1c, atp5f1c, atp5f1d, atp5f1e, atp5mc1, atp5mc2, atp5me, atp5mf, atp5mg, atp5pb, atp5pd, atp5pf, atp5po, atp6ap1, atp6v0a1, atp6v0a2, atp6v0a2, atp6v0c, atp6v0d1, atp6v0d2, atp6v0e1, atp6v1a, atp6v1b2, atp6v1c1, atp6v1c2, atp6v1d, atp6v1e1, atp6v1f, atp6v1g1, atp6v1h, cox15, cox17, cox4i1, cox5a, cox5b, cox6a1, cox6b1, cox6c, cox7a2, cox7a2l, cox7a2l2, cox7b, cox7c, cox8a, cox8b, cyc1, lhpp, ndufa1, ndufa10, ndufa10l1, ndufa11, ndufa12, ndufa13, ndufa2, ndufa4, ndufa5, ndufa6, ndufa7, ndufa8, ndufa9, ndufab1, ndufb10, ndufb11, ndufb2, ndufb3, ndufb4, ndufb5, ndufb6, ndufb7, ndufb8, ndufb9, ndufc2, ndufs1, ndufs2, ndufs3, ndufs4, ndufs5, ndufs6, ndufs7, ndufs8, ndufv1, ndufv2, ndufv3, ppa1, ppa2, sdha, sdhb, sdhc, sdhd, tcirg1, uqcr10, uqcr11, uqcrb, uqcrc2, uqcrfs1, uqcrh, uqcrq*	1.3 × 10^−8^
(3) C-650PND vs. C-110 PND Aging effect in controls	*atp5f1c, atp5mc1, atp5me, atp5mf, atp5mg, atp5pb, atp5pd, atp5pf, atp5po, atp6v0a2, atp6v0c, atp6v0d1, atp6v0e1, atp6v1a, atp6v1f, atp6v1g1, atp6v1h, cox15, cox17, cox4i1, cox5a, cox5b, cox6a1, cox6b1, cox6c, cox7a2, cox7a2l, cox7a2l2, cox7b, cox7c, cox8a, ndufa1, ndufa10l1, ndufa11, ndufa12, ndufa13, ndufa2, ndufa4, ndufa5, ndufa6, ndufa7, ndufa8, ndufa9, ndufb10, ndufb11, ndufb2, ndufb3, ndufb4, ndufb5, ndufb6, ndufb7, ndufb9, ndufs3, ndufs5, ndufs6, ndufs8, ndufv1, ndufv2, ndufv3, ppa1, ppa2, sdhb, sdhc, sdhd, tcirg1, uqcr10, uqcrb, uqcrfs1, uqcrh, uqcrq*	2.42 × 10^−14^	** *cox6a, cox8, atpev1c* **	---
(4) MO-650PND vs. MO-110PND Aging effect in MO	** *Cox8* **	---	*ap2s1, apaf1, atp5f1c, atp5f1d, atp5f1e, atp5mc1, atp5mc2, atp5pb, atp5pd, atp5pf, atp5po, bax, casp3, cox4i1, cox5a, cox5b, cox6a1, cox6b1, cox6c, cox7a2, cox7a2l, cox7a2l2, cox7b, cox7c, cox8a, cox8b, crebbp, creb3l1, cycs, cyct, cyct, dlg4, dnah1, gpx1, hdac2, ndufa1, ndufa10, ndufa11, ndufa12, ndufa13, ndufa2, ndufa4, ndufa5, ndufa6, ndufa7, ndufa9, ndufab1, ndufb10, ndufb11, ndufb2, ndufb3, ndufb4, ndufb5, ndufb6, ndufb7, ndufb9, ndufc2, ndufs3, ndufs4, ndufs5, ndufs6, ndufs7, ndufs8, ndufv2, ndufv3, plcb1, polr2f, polr2g, polr2h, polr2i, polr2j, polr2k, pparg, sdhd, slc25a5, sod1, sod2, uqcr10, uqcr11, uqcr11, uqcrb, uqcrfs1, uqcrh, uqcrq, vdac3*	<2.2 × 10 ^−16^

The genes in bold were up-regulated between comparisons in each sex.

## Data Availability

The data that support the findings of this study are openly available in NCBI’s Gene Expression Omnibus and are accessible through GEO Series accession numbers GSE115535 and GSE160153.

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
