# Peer review of "Maternal Obesity Programs the Premature Aging of Rat Offspring Liver Mitochondrial Electron Transport Chain Genes in a Sex-Dependent Manner"

_biology, 2023, doi:10.3390/biology12091166_

Round 1
Reviewer 1 Report
This is a well designed study that presents details on how maternal obesity can influence the hepatic gene expression of here offspring as well as influence how the liver changes as the offspring ages. The authors have presented an interesting study that adds to evidence from other groups that maternal diet during pregnancy can influence offspring metabolism, but goes beyond previous studies by specifically identifying sex-specific and age-specific hepatic changes.
Overall I enjoyed this manuscript, it would benefit from some minor amendments listed below with specific comments.
Overall comment: The nomenclature for the groups is confusing. The F1 is unnecessary as all comparisons are being made in the F1 generation. To make it easier to follow the authors could remove F1
The labeling of groups is different in almost all figures. For example, the images in Figure 5 are differently labeled from the graphs, and further different to Figure 4. Authors need to double-check all labels on all figures.
Figure 1: I don't agree that the box above the right hand Venn circle should say 'Natural aging' as (if the box below this circle is correct) this is the Maternal diet effect in F1 aged mice?
Also, the * is missing on the female genes in common overlap
Figure 3-7: Add negative representations for each stain. Data should be presented as dot plots with mean and SEM rather than bar charts - this gives a better representation of the spread of the data. Layout and size of representative images should be the same for the different sexes showing the same genes, and also ensure labels match (see overall comment previously).
Line 23: More DEGs compared to females?
Line 68: A bit jarring to use F1, would prefer F1 offspring or just offspring. Also, don't need F1 at end of sentence and beginning
Line 127: what does this mean. 12:00-14:00 would be during light period as previously stated?
Line 130: This is unclear. Do the authors mean that on 110 PND one male and one female per litter were euthanized and then at 650 PND the remaining siblings were euthanized?
Line 140/141: Unnecessary repetition
Line 191: Negative images should be uploaded to supplemental
Line 207-211 Clearly outlines the comparisons very nicely. It's a good addition
Line 212-221: This data would be better presented as a table so it is instantly apparent for each of the comparisons the number of up/down DEGs in males and females respectively. % could also be added to this table
Line 224-233: These statements don't match the comparisons stated in the Venn diagram? In Figure 1 it states the comparison is between F1MO-F1C 110PND and F1MO-F1C 650PND, in the box below the circles. This is not what is stated in the text. Please alter accordingly
Line 232-233: This statement doesn't make sense. Need to state a directional change i.e. affects more genes involved in premature aging in F1 males compared to F1 females?
Line 237: Please identify the significantly DEGs in a supplementary table.
Line 249-250: "To corroborate that the pathways involved in the comparison of F1MO-110PND vs F1C-110PND and F1C-650PND vs F1C-110PND, were also enriched and significant separately," does not make sense, please amend
Line 262: Don't need the word 'genes' after DEG
Line 315: Should be exhibiting
Line 342: F1MO-110PND doesn't have less expression than F1C-110PND (Fig7A)
Line 398-401: This statement should be rephrased to state what you found rather than what you studied.
Line 416: Delete 'in the mother' it is unnecessary repetition
Line 536-537: Not sure you can state it occurs in females but at older ages, as you only looked at 650PND in both. This needs rephrasing to avoid confusion
Line 539: The statement 'which was increased' is ambiguous. Please clarify what was increased.
Well-written with only minor errors, particularly with use of F1 instead of offspring. It should be F1 offspring or F1 generation not just F1 on it's own. Just using offspring after stating once that it is the F1 offspring would be fine in this manuscript.
Author Response
This is a well-designed study that presents details on how maternal obesity can influence the hepatic gene expression of her offspring as well as influence how the liver changes as the offspring ages. The authors have presented an interesting study that adds to evidence from other groups that maternal diet during pregnancy can influence offspring metabolism but goes beyond previous studies by specifically identifying sex-specific and age-specific hepatic changes. Overall, I enjoyed this manuscript, it would benefit from some minor amendments listed below with specific comments.
We are pleased to know that the reviewer enjoyed the manuscript, and we appreciate the reviewer's positive comments and suggestions for improvement.
The nomenclature for the groups is confusing. The F1 is unnecessary as all comparisons are being made in the F1 generation. To make it easier to follow the authors could remove F1.
We agree with the reviewer. F1 was removed from the group's nomenclature.
The labeling of groups is different in almost all figures. For example, the images in Figure 5 are differently labeled from the graphs, and further different to Figure 4. Authors need to double-check all labels on all figures.
We apologize for the mistake. We have now labeled the graphs consistently.
Figure 1: I don't agree that the box above the righthand Venn circle should say 'Natural aging' as (if the box below this circle is correct) this is the Maternal diet effect in F1 aged mice? Also, the * is missing on the female genes in common overlap.
The reviewer's comment helped us identify an error in the description of the compartments in Figure 1 (Venn Diagram). We sincerely apologize for the mistake which has been corrected. As a result, it is accurate to say that it is natural aging, given the comparison of controls at both ages. As suggested by the reviewer, the asterisk has now been included.
Figure 3-7: Add negative representations for each stain. Data should be presented as dot plots with mean and SEM rather than bar charts - this gives a better representation of the spread of the data. Layout and size of representative images should be the same for the different sexes showing the same genes, and also ensure labels match (see overall comment previously).
Negative representations for each stain have now been included as supplementary material.
We appreciate your suggestion, however due to the modest dispersion observed, we prefer to provide the data in its current form (bar chart).
We have now labeled the representative images consistently.
Line 23: More DEGs compared to females?
We revised and corrected the information.
Line 68: A bit jarring to use F1, would prefer F1 offspring or just offspring. Also, don't need F1 at end of sentence and beginning.
We appreciate the feedback and have modified the sentence.
Line 127: what does this mean. 12:00-14:00 would be during light period as previously stated?
The tissue collection was performed during the light period. The text has been revised and changed.
Line 130: This is unclear. Do the authors mean that on 110 PND one male and one female per litter were euthanized and then at 650 PND the remaining siblings were euthanized?
One male and female F1 per litter were euthanized by exsanguination through aortic puncture under isoflurane general anesthesia at each age, 110 and 650 PND by the same experienced person under identical conditions (light period (12:00 to 14:00 h) and 6 h of fasting).
Line 140/141: Unnecessary repetition.
We agree with the reviewer, and we have deleted the repeated information.
Line 191: Negative images should be uploaded to supplemental.
The negative images are now included as additional material.
Line 207-211 Clearly outlines the comparisons very nicely. It's a good addition.
We appreciate your suggestion. We have tried to make the comparisons clearer.
Line 212-221: This data would be better presented as a table so it is instantly apparent for each of the comparisons the number of up/down DEGs in males and females respectively. % could also be added to this table.
We prefer to keep these data in text.
Line 224-233: These statements don't match the comparisons stated in the Venn diagram? In Figure 1 it states the comparison is between F1MO-F1C 110PND and F1MO-F1C 650PND, in the box below the circles. This is not what is stated in the text. Please alter accordingly.
The statement in the text is correct, however the labels in figure 1 (Venn diagram) were incorrect. We apologize for the mistake and have corrected the figure.
Line 232-233: This statement doesn't make sense. Need to state a directional change i.e. affects more genes involved in premature aging in F1 males compared to F1 females?
Thank you for the observation. We checked and corrected the sentences.
Line 237: Please identify the significantly DEGs in a supplementary table.
As suggested by the reviewer, a supplementary table containing the DEGs has been included.
Line 249-250: "To corroborate that the pathways involved in the comparison of F1MO-110PND vs F1C-110PND and F1C-650PND vs F1C-110PND, were also enriched and significant separately," does not make sense, please amend.
Thank you for the observation. We checked and corrected.
Line 262: Don't need the word 'genes' after DEG.
Thank you for the observation. We have now omitted the word genes.
Line 315: Should be exhibiting.
We appreciate your observation. The requested modification has been incorporated.
Line 342: F1MO-110PND doesn't have less expression than F1C-110PND (Fig7A).
We appreciate your observation. The modification has been incorporated.
Line 398-401: This statement should be rephrased to state what you found rather than what you studied.
We appreciate your observation. The requested modification has been incorporated.
Line 416: Delete 'in the mother' it is unnecessary repetition.
Thank you for the observation. We have removed the unnecessary repetition.
Line 536-537: Not sure you can state it occurs in females but at older ages, as you only looked at 650PND in both. This needs rephrasing to avoid confusion.
We agree with the reviewer and have now state that: ”in males at 110 PND, maternal obesity accelerates the age-associated downregulation of genes and pathways related to mitochondrial function. In females, these programming effects occur at 650 PND”.
Line 539: The statement 'which was increased' is ambiguous. Please clarify what was increased.
We appreciate the feedback and have modified the sentence.
Reviewer 2 Report
The authors investigate the effects of maternal obesity on rat offspring liver mitochondrial function in a sex-specific manner.
In general the manuscript was well written and study design was good.
Comments:
Line 19: add age for young (PND110) and old (PND650)
Line 40 needs a reference
I think that removal of the term F1 and replacing this with the term offspring is appropriate. Also removal of the term F0 and just replace with maternal/mother
Line 112: There needs to be a brief acknowledgement that adjustment of litter sizes may impact results https://pubmed.ncbi.nlm.nih.gov/27689313/
The authors should be explicit that only 1 male and 1 female from each litter and each timepoint were studied. Littermates are replicates
Table 3 and Figure 2 are difficult to read-can they be expanded
Fig 3I, 4I-scale bar needed (only appears in NDUFA10 )
Line 90- Minor formatting errors eg ac-cordance
Author Response
The authors investigate the effects of maternal obesity on rat offspring liver mitochondrial function in a sex-specific manner. In general, the manuscript was well written and the study design was good.
We are pleased to know that the reviewer found the manuscript to be well-written and well-designed. Additionally, we appreciate the reviewer's positive remarks and suggestions for improvement.
Comments:
Line 19: add age for young (PND110) and old (PND650).
As suggested by the reviewer, the ages for young and old were added.
Line 40 needs a reference.
As suggested a reference was included.
I think that removal of the term F1 and replacing this with the term offspring is appropriate.
Since the presented data relate directly to the offspring, we decided to exclude F1 from the group designations.
Also removal of the term F0 and just replace with maternal/mother.
As suggested the term F0 was replaced.
Line 112: There needs to be a brief acknowledgement that adjustment of litter sizes may impact results https://pubmed.ncbi.nlm.nih.gov/27689313/.
Manipulating litter size impacts pups’ postnatal growth and development. However, these effects are typically observed when litter size is altered to 3-5 pups per litter (overnutrition) or 16-18 pups per litter (undernutrition). As suggested, we reviewed Dickinson's manuscript; predominantly, when discussing manipulation of litter size, the authors cite to articles in which litter sizes (9–14 rats/litter) were lowered to 3-5 pups/litter.
According to our experience, the average litter size at birth can range from 8 to 18 pups; therefore, the most common litter size adjustment we employ is 10 or 12 pups. This litter size adjustment has no effect on the metabolic variables as the litter size is considered normal. Usually, for litter adjustment, we try to ensure similar conditions during pregnancy; thus, in this study, litters with less than 9 or more than 14 pups were excluded. This has now been included in the text.
The authors should be explicit that only 1 male and 1 female from each litter and each timepoint were studied. Littermates are replicates.
As also suggested by reviewer 1 we have now clarified that only 1 male and 1 female from each litter and each time point were studied.
Table 3 and Figure 2 are difficult to read-can they be expanded.
We modified table 3 and figure 2 to make it clear.
Fig 3I, 4I-scale bar needed (only appears in NDUFA10).
Thanks for your observation, we have now included the scale bar.